# Highly Active Co_3_O_4_-Based Catalysts for Total Oxidation of Light C1–C3 Alkanes Prepared by a Simple Soft Chemistry Method: Effect of the Heat-Treatment Temperature and Mixture of Alkanes

**DOI:** 10.3390/ma14237120

**Published:** 2021-11-23

**Authors:** Rut Sanchis, Adrián García, Francisco Ivars-Barceló, Stuart H. Taylor, Tomás García, Ana Dejoz, María Isabel Vázquez, Benjamín Solsona

**Affiliations:** 1Departamento de Ingeniería Química, Universitat de València, C/Dr. Moliner 50, Burjassot, 46100 Valencia, Spain; rut.sanchis@uv.es (R.S.); adrian.garcia@uv.es (A.G.); ana.m.dejoz@uv.es (A.D.); isabel.vazquez@uv.es (M.I.V.); 2Departamento Química Inorgánica y Química Técnica, Faculty of Sciences, UNED, Av. Esparta s/n, Las Rozas, 28232 Madrid, Spain; 3Cardiff Catalysis Institute, School of Chemistry, Cardiff University, Main Building, Park Place, Cardiff CF10 3AT, UK; taylorsh@cardiff.ac.uk; 4Instituto de Carboquímica (ICB-CSIC), C/Miguel Luesma 4, 50018 Zaragoza, Spain; tomas@icb.csic.es

**Keywords:** cobalt oxide, total oxidation, alkane oxidation, heterogeneous catalysis, volatile organic compounds, propane, ethane, methane, surface oxygen vacancies

## Abstract

In the present work, a simple soft chemistry method was employed to prepare cobalt mixed oxide (Co_3_O_4_) materials, which have shown remarkably high activity in the heterogeneously catalyzed total oxidation of low reactive VOCs such as the light alkanes propane, ethane, and methane. The optimal heat-treatment temperature of the catalysts was shown to depend on the reactivity of the alkane studied. The catalytic activity of the Co_3_O_4_ catalysts was found to be as high as that of the most effective catalysts based on noble metals. The physicochemical properties, from either the bulk (using XRD, TPR, TPD-O_2_, and TEM) or the surface (using XPS), of the catalysts were investigated to correlate the properties with the catalytic performance in the total oxidation of VOCs. The presence of S1 low-coordinated oxygen species at the near surface of the Co_3_O_4_-based catalysts appeared to be linked with the higher reducibility of the catalysts and, consequently, with the higher catalytic activity, not only per mass of catalyst but also per surface area (enhanced areal rate). The co-presence of propane and methane in the feed at low reaction temperatures did not negatively affect the propane reactivity. However, the co-presence of propane and methane in the feed at higher reaction temperatures negatively affected the methane reactivity.

## 1. Introduction

The elimination of volatile organic compounds (VOCs) is an extremely important issue, as the release of VOCs into the environment has serious implications. Pollution by VOCs is related to the formation of photochemical smog, the depletion of tropospheric ozone as well as to a wide range of human diseases (many of them are toxic and/or carcinogenic). Reduction of VOC emissions can be addressed through several options such as incineration, catalytic removal, adsorption, absorption, condensation, and biofiltration. Comparing the options, the catalytic removal by total oxidation is one of the most efficient methods [1,2,3,4,5,6] and is especially effective for treating low VOC concentrations released from continuous processes [7,8].

Commercial catalysts for VOC total oxidation are, in most cases, based on noble metals, such as platinum or palladium, due to the fact of their high catalytic activity [9,10,11,12,13,14]. Metal oxides of non-noble metals are usually less active but have the advantage of presenting a comparatively lower cost [7,15]. Several studies have investigated transition metal oxide catalysts and demonstrated that oxides of manganese [16,17,18] or cobalt are also highly active [19,20,21,22,23]. In fact, Co_3_O_4_ presents a catalytic activity for total oxidation of alkanes (the least reactive VOCs) comparable or even greater than those of noble metal- based catalysts [24,25,26]. 

Cobalt oxide in its spinel Co_3_O_4_ form presents Co^3+^ and Co^2+^ ions in octahedral and tetrahedral sites, respectively [27], with a mean cobalt oxidation state of +2.67. The high catalytic activity of Co_3_O_4_ can be closely related to a series of characteristics: (i) being a p-type semiconducting metal oxide, (ii) its spinel phase has the highest oxygen binding rate [28] with a very low oxygen bond strength [29], and (iii) cobalt atoms existing in two different oxidation states within the same phase. These features are especially important for facilitating oxygen migration and providing versatile and powerful redox properties.

The oxidation mechanism of saturated hydrocarbons, such as propane, is still a matter of debate, and there are several points of view. The oxidation mechanism on p-type oxide catalysts, such as Co_3_O_4_, has been reported to need the VOC to strongly adsorb on an anionic oxygen site from the metal oxide lattice [3,30,31]. Consequently, an activated complex is formed on the catalyst surface, which reacts with molecular oxygen to finally yield carbon dioxide.

Other works propose that the catalytic activity of the cobalt oxide is controlled by the dissociative adsorption of molecular oxygen [28,32]. Nevertheless, the role of the lattice oxygen in the catalytic reaction appears still to be essential [33,34] as well as the presence of the reactive {112}/{110} planes from the Co_3_O_4_ structure [4]. The role of the lattice oxygen as active species and the Mars–van Krevelen redox mechanism in the total oxidation of methane and propane using Co_3_O_4_ catalysts was confirmed using a pulse reactor by feeding labeled ^18^O_2_ oxygen [35,36]. Thus, only C^16^O_2_ was initially detected in the hydrocarbon total oxidation, indicating that the oxygen required for the CO_2_ formation comes exclusively from the catalyst. Neither C^18^O_2_ nor C^18^O^16^O products were initially detected.

Interestingly, it seems that the morphology of cobalt oxide also plays an important role. Thus, Co_3_O_4_ nanosheets have shown higher reactivity in methane oxidation than Co_3_O_4_ nano-sized belts or cubes [37], in spite of the fact that nanosheets have presented lower surface area. Thus, it seems that {112} exposed planes of nanosheets are more reactive than the planes exposed in the other morphologies. Similarly, Fei and coworkers prepared Co_3_O_4_ nanotubes showing very high catalytic activity and good stability in methane oxidation also relating this behavior to the presence of {112} exposed planes [38]. Nanotubes were more active than Co_3_O_4_ nanorods and nanoparticles. The reactivity of the {112} planes has been associated with their open structure and, consequently, to the low electronic energy barrier for C–H bond breaking compared with other more packed planes [4,37].

Co_3_O_4_ presents a relatively high stability in catalytic total oxidation under conventional conditions, as long as the reaction temperatures employed are not relatively high [39,40]. However, the thermal stability of Co_3_O_4_ is destabilized at increasing reaction temperatures as well as when using high-temperature activation treatments. In this way, activation temperatures over 700 °C have been proposed to be responsible for sintering, changes in the surface of the catalysts, and transformation of Co_3_O_4_ into the less active CoO phase among other effects, leading to a lack of stability for cobalt oxide-based catalysts [41,42,43,44,45].

A high efficiency for the removal of chlorinated VOCs by total oxidation has also been reported for cobalt-oxide-derived catalysts [46,47]. In these articles, the catalytic activity for the oxidative degradation of chlorinated compounds was observed to be higher for the smaller Co_3_O_4_ crystallites, which was justified by having more easily accessible active sites (O^−^ or O^2−^ species) in contrast with Co_3_O_4_ catalysts presenting higher crystallinity. In the same vein, a better dispersion of cobalt oxide, along with a higher reducibility, was reported to be responsible for an enhanced catalytic activity in toluene total oxidation [47,48,49]. On the other hand, the removal of VOCs dissolved in the liquid phase has been also described using catalysts based on bulk Co_3_O_4_ or supported on activated carbons [50]. In this case, the activation of the VOC (toluene) was proposed to proceed via a different mechanism than in the gas phase, and Co^−^OH^+^ and highly dispersed Co_3_O_4_ species were identified as active sites.

In this work, we present an extremely simple and straightforward method to produce cobalt oxides with high catalytic activity in the total oxidation of propane, ethane, and even methane as model compounds, which could be present in the exhaust gases of different combustion processes, especially those using liquid petroleum gas as fuel. Moreover, the calcination temperature for the final step of the catalyst preparation is approached and discussed as a key parameter for properly activating the catalytic properties of these materials for the reactions at issue. Then, the optimal calcination temperature depends on the reactivity of the alkane fed. Finally, the effect of the use of mixtures of alkanes was also studied.

## 2. Materials and Methods

### 2.1. Synthesis of Co_3_O_4_-Based Catalysts

Oxalic acid was added to an aqueous solution of cobalt (II) nitrate (oxalic acid/Co molar ratio of 3). The mixture was evaporated at 80 °C until a slurry was obtained. The resulting mixture was dried overnight at 120 °C and calcined in static air at different temperatures (over the range 300–750 °C) for 3 h. The catalysts were named as CoX, X being the calcination temperature employed.

### 2.2. Characterization Techniques

The specific surface areas of the catalysts were determined by the Brunauer–Emmett–Teller (BET) model from N_2_ adsorption–desorption isotherms measured at −196 °C using a Micromeritics ASAP-2020 automated analyzer. Prior to the analyses, the catalysts were degassed in vacuum (10^–6^ Torr) at 80 °C for 5 h.

Powder X-ray diffraction (XRD) patterns were collected with a Bruker D8 Advance Series 2 diffractometer, employing a Cu Kα source (λ = 1.5418 Å) operated at 40 kV and 40 mA.

Compositional, morphological, and structural analyses of the catalysts were carried out by high-resolution transmission electron microscopy (HRTEM) using a field emission gun TECNAI G2 F20 microscope operated at 200 kV, equipped with electron diffraction (ED) and energy-dispersive X-ray spectroscopy (EDX) to perform selected area (SA) analyses: SA-ED and SA-EDX, respectively. The samples for HRTEM were prepared by sonication of the powders in pure ethanol for several minutes. Then, a drop of the resulting suspension was placed on a holey carbon film (in turn, supported on a copper grid) and the solvent was dried at room temperature.

X-ray photoelectron spectra (XPS) were acquired using a SPECS spectrometer equipped with a PHOIBOS 100 MCD detector and monochromatic Al Kα (hv = 1486.6 eV) X-ray radiation operating at 300 W (15 keV and 20 A). The default detector voltage was kept constant at 2100 eV for all measurements, and the spectra were acquired with a pass energy of 25 eV and an energy step of 0.2 eV. The base pressure in the analysis chamber was kept below 5 × 10^−9^ mbar. XPS peak areas were calculated integrating individual components obtained by subtracting the Shirley-shape background. Spectrometer transmission function, cross-section, and inelastic mean free path values from CasaXPS software were used for quantitative calculations.

Temperature programmed reduction (TPR) analyses were carried out in a Micromeritics Autochem 2920 instrument equipped with a thermal conductivity detector, operated under a 50 mL min^−1^ 10% H_2_/Ar flow at temperatures between 50 and 800 °C, with a heating rate of 10 °C min^−1^.

### 2.3. Catalytic Tests

Catalytic activity was determined employing a fixed bed laboratory microreactor. For each catalytic test, 0.2 g of catalyst was placed in a quartz tube reactor (1/200 o.d.). The feed of the reactor contained 8000 vppm of alkane (propane, ethane, or methane) in a mixture of O_2_ and He (O_2_/He molar ratio of 0.2/0.8) with a flow rate of 50 mL min^−1^. Additionally, some selected experiments were conducted with two alkanes together (methane and propane) feeding 8000 vppm of propane and 8000 vppm of methane, with the remaining O_2_ and He (O_2_/He molar ratio of 0.2/0.8) with a flow rate of 50 mL min^−1^.

The reactants and reaction products were analyzed using an online gas chromatograph equipped with TCD and FID detectors. Two chromatographic columns were used: a Porapak Q (for CO_2_, water, and hydrocarbons separation), and a Molecular Sieve 5A (for CO, O_2_, and N_2_ separation). The reaction temperatures used, within the 100–500 °C range, were measured by a thermocouple introduced into a pod placed in contact with the catalyst bed. Conversion data were determined through the differences between the inlet and outlet concentrations. The chromatographic area of CO_2_ was used as the comparative reference in order to corroborate the conversion data (carbon balance always adjusted with an accuracy of ±4%). The catalytic results were averaged from 4 repetitions of analyses conducted when steady-state activity was attained (after approximately 40 min.). Blank runs carried out from 100 to 500 °C in an empty reactor showed no activity.

## 3. Results

### 3.1. Heterogeneously Catalyzed Total Combustion of Light Alkanes

The catalytic performance of the cobalt-oxide-based catalysts, prepared using a soft chemistry method and activated by a heat-treatment at different temperatures, was studied in the total oxidation of light alkanes (C_1_–C_3_), which had reactivity that increased with the hydrocarbon chain length. Thus, starting with the most reactive alkane tested, the variation of propane conversion with the reaction temperature for the different catalysts (Figure 1), showed a higher catalytic activity as the calcination temperature used for the catalyst activation was lowered. Therefore, for the lowest calcination temperature of 300 °C, the resulting catalyst Co300 offered the highest catalytic activity, starting to convert propane from 150 °C and reaching 100% conversion at a temperature as low as 225 °C. In contrast, for the catalyst calcined at the highest temperature (Co750), full conversion was achieved at a markedly higher temperature, 350 °C. For an intermediate calcination temperature (Co550 catalyst), 100% propane conversion was obtained at 250 °C. Furthermore, several catalytic cycles of propane oxidation were undertaken with the two most active samples: the Co300 and Co550 catalysts. Interestingly, these catalysts proved to be stable, at least for the three catalytic runs tested up to the temperature of 100% propane conversion (Figure 1). We must note that CO_2_ was the main reaction product observed, the selectivity to CO_2_ being 100% in all cases at conversions over 5%. At lower conversions, traces of propylene and CO were also detected.

Additional catalytic tests using a less reactive alkane, ethane, and the least reactive of the alkanes, methane, were also undertaken with Co300 and Co550 catalysts, the ones showing the highest activity for propane. The reactivity results for ethane and methane total oxidation are comparatively displayed in Figure 2, along with the data obtained for propane at equivalent reaction conditions.

As expected, the longer the hydrocarbon chain, the lower the reaction temperature to reach alkane iso-conversion for a given catalyst. Thus, methane required the highest reaction temperature for both catalysts, followed sequentially by ethane and then propane (Figure 2). In agreement with the results initially obtained for propane, the lower the calcination temperature used for the catalyst activation, the higher the catalytic activity shown for the hydrocarbon total oxidation. Thus, the same trend was observed for the total oxidation of all three alkanes. As a summary for all the catalysts and alkanes tested, the corresponding reaction temperatures to reach 20% alkane conversion are presented in Table 1.

For the case of methane, the smallest and the most stable alkane, its low reactivity did not allow us to achieve a total conversion at a reaction temperature lower than 300 °C, which should not be exceeded with the Co300 catalyst, as it was the calcination temperature used in the activation treatment. On the other hand, it is a noteworthy fact that the Co550 catalyst was able to achieve approximately 100% methane conversion at a temperature as low as 450 °C, far below the calcination temperature employed for its activation (Figure 2).

### 3.2. Catalyst Calcination Temperature: Effects on Structural, Redox, and Surface Features

The surface area of the Co_3_O_4_ catalysts calcined at different temperatures was determined by BET adsorption isotherms with N_2_, and the results show that the higher the calcination temperature used, the lower the specific surface area for the resulting catalyst. Thus, a calcination temperature of 300 °C, the lowest used, gave rise to the catalyst (Co300) presenting the highest surface area of 65.0 m^2^ g^−1^, which decreased approximately three times for the catalysts resulting from a 550 °C calcination (Co550), with 22.1 m^2^ g^−1^. Finally, the surface area drastically reduced to 2.1 m^2^ g^−1^ for the Co750 catalyst, calcined at the highest temperature (750 °C).

XRD patterns confirm that only reflections consistent with the Co_3_O_4_ crystalline phase (JCPDS: 42-1467) were observed in the catalysts calcined at 300, 550, and 750 °C (Figure 3) at 2θ of 19.2, 31.4, 37.0, 45.0, 55.8, 59.5, and 65.3° [39].

Further analysis of the XRD peaks by the Scherrer equation show important variations in the average size of the crystallites formed in each catalyst depending on the calcination temperature used for their activation. In this sense, the Co_3_O_4_ crystallite size increased with the calcination temperature according to the following trend: Co300 (20 nm) < Co550 (32 nm) < Co750 (56 nm), which is coherent with the reverse increase in surface areas (Table 1).

It must be said that the Scherrer equation was applied assuming a scattering vector of 0.9, conventionally used for globular particle morphologies with cubic symmetry [51]. Indeed, characterization by high-resolution transmission electron microscopy (HRTEM) confirmed a relatively globular-like morphology for the Co_3_O_4_ crystalline basic units for all three samples calcined at 300, 550, or 750 °C (Figure 4). In the sample calcined at 300 °C (Co300), porous rod-shaped superstructure domains were also observed.

These 3D superstructures can, in some cases, reach a length of 4–5 μm and appear to consist of much smaller interconnected cobalt oxide nanoparticles with diameters mainly within the 5–25 nm range. Unfortunately, a size distribution analysis was not possible due to the complication of the overlapping nanoparticles in all space directions. In contrast, the sample calcined at 550 °C presented a very distinct morphology, and it was formed of isolated and randomly aggregated nanoparticles with a mean size of 34.3 nm (standard deviation of ±6.4 nm), calculated by direct measurements on the HRTEM micrographs of Co550 catalyst, in good agreement with that obtained by the Scherrer equation from the XRD data (Table 1). Finally, the sample heat-treated at 750 °C (Co750) showed a morphology very similar to that of Co550, but with markedly larger particles with diameters mainly within the 34–180 nm range, which indicates a very broad size distribution.

Further HRTEM analysis by selected area electron diffraction (SAED) showed a diffraction pattern with six well-defined diffraction rings that can be indexed as a Co_3_O_4_ cubic symmetry structure (JCPDS: 42-1467) with space group Fd3m (inset Figure 4). The interplanar distances (d-values), determined from the radius of the rings at 4.68, 2.88, 2.45, 2.04, 1.67, and 1.55 Å (±0.02 Å), corresponded to the lattice planes {111}, {220}, {311}, {400}, {422}, and {511}, respectively. The concentric diffraction rings instead of diffraction spots are characteristic of a polycrystalline sample, that is, a mixture of cobalt oxide nanoparticles randomly distributed and oriented, resulting in a continuous angular distribution of (h, k, and l) spots so close to each other that they form rings.

It is important to note that no further diffraction rings related to any other crystalline phase were observed in the SAED patterns collected, which confirms the nanostructured catalysts exclusively consisted of pure Co_3_O_4_ phase. Additionally, the local chemical composition determined by EDX on areas selected by HRTEM in nanoprobe mode (beam spot size < 5 nm) on single nanoparticles was consistent with a Co_3_O_4_ stoichiometry.

Furthermore, high-resolution TEM images of the Co550 catalyst, displayed in Figure 5, showed no apparent presence of multiple grains oriented in different crystallographic directions within a single nanoparticle. Instead, a well-ordered atomic lattice with lattice fringes clearly defined was observed, in agreement with monocrystalline Co_3_O_4_ nanoparticles and similar to what was observed for the HRTEM micrograph of Co300 catalyst in Figure 4.

Direct spacing measurements of lattice fringes from an HRTEM micrograph (Figure 5a) gave rise to 4.67 Å and 2.44 Å values which corresponded to the {111} and {311} lattice spacings of Co_3_O_4_, respectively. A fast Fourier transformation (FFT) (Figure 5b) of the selected nanoparticle unambiguously confirmed the presence of the crystal planes identified as well as the monocrystalline structure of the Co_3_O_4_ nanoparticle. Interestingly, the lattice parameter (lattice constant a) calculated for all the fresh catalysts from the interplanar distances, observed by HRTEM and FFTs, was similar, not dependent on the chosen direction (isotropic), and consistent with the reported data for a Co_3_O_4_ cubic spinel structure [52]: 0.810 nm for the Co300 catalyst and 0.811 nm for both the Co550 and Co750 catalysts.

Beyond the particle size or the structural and textural properties, the different calcination temperatures induced significant changes in the redox properties of these Co_3_O_4_-based catalysts. Thus, the reducibility of the cobalt oxide sites decreased with the catalyst calcination temperature as observed from the temperature-programmed reduction experiments with H_2_ (TPR-H_2_) conducted on the Co300, Co550, and Co750 samples. Although a general shift towards higher temperatures for maximum reduction rates was observed when increasing the calcination temperature, a similar profile was maintained for all three catalysts (Figure 6), presenting two main peaks with low and higher intensities, overlapping to a greater or lesser extent at lower and higher reduction temperature, respectively. The maximum for the low-intensity reduction peak appeared within the 238–340 °C temperature range, while the high-intensity one was observed at higher temperatures from 321 to 408 °C, corresponding to the Co_3_O_4_ to CoO and CoO to Co reductions steps, respectively, according to the reported main steps for the Co_3_O_4_ reduction [43]. However, some other intermediate compounds must form during the reduction for these samples, as a clear shoulder was observed before the maximum within the highest temperature reduction peak. In the case of the sample heat-treated at 300 °C, a TPR-H_2_ with peaks at 238 and 321 °C and a shoulder at 270 °C was observed. Additionally, a low-intensity reduction peak can be detected at low temperatures, with the onset below 100 °C and the maximum at 138 °C, which might be caused by cobalt nitrate impurities from the catalyst synthesis process and/or by a hydrogen spillover effect [53,54]. The presence of surface adsorbed oxygen species could be also related to this low temperature reduction peak. On the other hand, the Co550 catalyst presented two main reduction peaks at approximately 280 (low intensity) and 350 °C (high intensity) and a shoulder at approximately 310 °C. Finally, the reduction maxima for the Co750 catalyst appeared at 336 and 408 °C, with no distinguishable shoulder.

The differences observed in the reducibility of the Co_3_O_4_-based catalysts might be partially induced by variations in the nanometric particle size. Nevertheless, an additional correlation was found with the near-surface chemical composition of cobalt oxide catalysts studied by XPS. On the one hand, the quantitative analysis of the Co2p core-level signals (Figure 7a) showed no meaningful differences among all three catalysts, with a Co^2+^/Co^3+^ ratio of approximately one in all cases, implying a near-surface mean cobalt oxidation state of 2.5. The fact that this near-surface ratio was far higher than the theoretical one of 0.5 corresponding to the Co_3_O_4_ phase (mean cobalt oxidation state of 2.67), suggests a significant presence of surface oxygen vacancies.

The detailed analysis of the O1s core-level XPS signal (Figure 7b) evidenced the presence of two different types of oxygen with a different concentration depending on the catalyst. The region with a binding energy between 529 and 530 eV was characteristic of lattice oxygen, denoted as Oα, which has been shown to increase with the calcination temperature of our Co_3_O_4_-based catalysts (Table 2). The region between 531 and 533 eV has been related to non-stoichiometric oxygen species, denoted as Oβ, in the near-surface region of mixed cobalt oxides [55,56,57,58]. Within the latter region, three different oxygen species could be identified with peaks centered at 530.6, 531.6, and 532.9 eV, denoted as S1, S2, and S3 species, respectively [52,53,54,55]. The first two (i.e., S1 and S2) were associated with structural defects and/or low-coordinated surface oxygen, while S3 was related to the presence of hydroxyl species, which were hardly present in our catalysts (≤5% of total near-surface oxygen). The amount of S3- and S2-type oxygens remained unchanged in all three catalysts. Only the amount of S1 species underwent variations, increasing as the calcination temperature decreased for the catalysts developed.

### 3.3. Correlation between Catalytic Performance and Physicochemical Features

New experiments at a fixed reaction temperature (190 °C) were undertaken in propane oxidation with these catalysts. One hundred and ninety degrees Celsius was selected, as at this reaction temperature, all the catalysts could activate propane, but the alkane conversion did not exceed 25%.

According to these experiments, the reactivity data can be explained by considering either the catalytic activity normalized per gram of catalyst (Figure 8a) or per surface area (Figure 8b) of catalysts calcined at different temperatures. Firstly, the increase in the calcination temperature brought about a change in the textural properties of the catalysts, associated with a higher crystallite size and a lower surface area, which justified the decrease in the absolute catalytic activity (Figure 8a). However, if the surface area is considered, a drop in the specific activity, areal rate, was also observed (Figure 8b). In the latter case, the partial activity lost must be related to the lower reducibility found for the co-sites in the catalysts with larger Co_3_O_4_ particles. A higher reducibility was observed with a decreasing calcination temperature, in turn, linked to a higher number of near-surface structural defects (S1 oxygen species). Nevertheless, higher transfer of reactive lattice oxygens species to the catalyst surface favored by a lower particle size cannot be discarded as also contributing to the higher catalytic activity for light alkane combustion.

Overall, we observed that the most selective catalyst was that heat-treated at the lowest temperature (Co300). However, this sample would only be useful for the elimination by total oxidation of propane but not of ethane or methane, since these molecules are far less reactive than propane. Conversely, a calcination temperature of 550 °C seemed to be the optimal one, as it is a trade-off between reactivity (due to the notable reducibility and high concentrations of S1 oxygen species) and stability (as it can remove all methane at temperatures (100% conversion) below 500 °C). A stability test was performed with this Co550 sample showing that at a reaction temperature of 225 °C, the propane conversion initially observed was maintained for more than 24 h. The catalyst after use was characterized by TEM, showing no appreciable difference in terms of crystallite size compared with the fresh sample. Next, we decided to study the robustness of the Co550 catalyst, by undertaking experiments with steam and, also, mixing hydrocarbons (Figure 9).

Figure 9a shows the effect of adding water (10 wt.%) to the feed, replacing a part of the helium diluter by water, in one experiment with propane. A slight decrease in the propane conversion was observed. Interestingly, after removing the steam from the feed, the catalytic performance was fully restored, demonstrating a high robustness.

In addition, we studied the total oxidation of mixtures of propane and methane at two different reaction temperatures (Figure 9b). We selected (i) 235 °C, since at this reaction temperature Co550 can easily activate propane but can hardly activate methane; and (ii) 350 °C, since at this temperature the methane molecule can be activated by Co550, whereas if fed alone, the propane would be completely (or almost) converted. Initially only propane was fed at 235 °C, leading to a conversion of approximately 80% which kept stable for approximately 2 h. Then, methane was fed alone leading to very low conversion levels. In the third stretch, methane was added together with propane. Meanwhile methane could hardly be converted at that reaction temperature, and the propane conversion slightly increased until approximately 85%. At this point, the reaction temperature increased until 350 °C and then only propane was fed, with the conversion of propane exceeding 95%. In the next step, methane alone was fed. A stable methane conversion of approximately 50% was obtained. In the final step, propane was co-fed together with methane. It was observed that propane was completely transformed into CO_2_, whereas methane conversion decreased compared to the level achieved without propane (approximately 50% without propane and 20% with propane). Therefore, we can conclude that the competitiveness of active sites in adsorbing and reacting propane and methane molecules did modify the reactivity of the cobalt active sites depending on the conditions. At a low reaction temperature (235 °C), propane was converted into CO_2_ and was not affected by the presence of methane, probably due to the low adsorption of methane on the surface of the catalysts at that temperature. However, at higher temperatures (350 °C), the presence of propane led to an important drop in the methane conversion, and this was likely due to the adsorption and reaction of propane on the cobalt oxide surface sites, decreasing the amount of available active sites for methane activation.

The catalytic results obtained in the present study for the total oxidation of alkanes were in the same order of those obtained by the best cobalt oxide catalysts reported in the literature. Then, in propane total oxidation, to reach a conversion of 90%, a reaction temperature of approximately 215 °C was required with our optimal catalyst. This result was similar to that achieved by an ordered high surface area Co_3_O_4_ synthesized by a nanocasting route [20]. In other recent studies, the influence of the morphology (and its relationship with other physicochemical properties) over the catalytic activity was studied. Using rods of Co_3_O_4_, a propane conversion of 90% was obtained at 195 °C [40]. In a similar study, Co_3_O_4_ was prepared through a solid-state method to reach a propane conversion of 50%, at reaction temperatures higher than 210 °C [39]. In another work, it was demonstrated that the catalytic activity of Co_3_O_4_ catalysts was facet-dependent, reaching 90% conversion at 239 °C in the best cases [25]. The effect of pH in the synthesis of cobalt oxide has been also studied on the catalytic performance in propane total oxidation. It was reported at a conversion of approximately 90% at 220 °C, with a Co_3_O_4_ catalyst synthesized using high pH [31].

On the other hand, in the total oxidation of methane to obtain conversions of 10% or 90% with Co550 in this study, temperatures of approximately 280 or 410 °C, respectively, were required. Similarly, using an optimized N-doped Co_3_O_4_ catalyst of 10% methane conversion was achieved at 300 °C, and to obtain a conversion of 90%, 400 °C were required [59].

As a summary, we can conclude that cobalt oxide Co_3_O_4_-based catalysts are highly active for the total oxidation of highly stable light alkane (C_1_–C_3_) VOCs when prepared by a simple method using oxalic acid. The calcination temperature of the precursor at 300 °C gave rise to the most active catalyst (i.e., Co300), which had a Co_3_O_4_ spinel phase. Calcinations at temperatures above 300 °C produced no significant change in the Co_3_O_4_ spinel structure, although an important decrease in both the catalytic activity per mass unit of catalyst and per surface area were exhibited. The decrease in activity was associated to an increase in the particle size and consequent decrease in the surface area, induced by increasing calcination temperatures. An additional influence of the variations in the superstructure morphology cannot be completely discarded. In this sense, the porous morphology of the most active catalyst, Co300, changed to isolated and light aggregates of randomly arranged nanoparticles for the Co_3_O_4_ catalysts calcined at higher temperature. On the other hand, the highest areal rates obtained by the sample calcined at 300 °C appeared most likely related to an enhanced lability of lattice oxygen, evidenced by the reducibility demonstrated by TPR experiments. Despite the severe decrease in the surface area, the calcination at 750 °C led to a lower catalyst reducibility, and this meant an additional drop in both the catalytic activity per mass of catalyst and per surface area. Calcination at 750 °C produced no significant variations in particles shape compared to that at 550 °C, only the formation of larger particles. A structural study by XRD and microscopy demonstrated that the lattice parameters remained very similar regardless of the calcination temperature in all three catalysts (i.e., Co300, Co550, and Co750).

As the oxidation of propane to CO_2_ on cobalt oxide is widely known to be proceeded by a redox mechanism, the decrease in the rate of oxygen supply (reducibility) associated with the increasing calcination temperature determined the lowest areal rate. Interestingly, the amount of S1 low-coordinated oxygen species at the near surface of the Co_3_O_4_-based catalysts appeared to be linked with the higher reducibility and, consequently, with the higher areal rate. The optimal calcination temperature depended on the alkane fed. Thus, if VOCs with high reactivity are tested, it is preferable the use low calcination temperatures (i.e., 300 °C). However, if VOCs with low reactivity, such as methane, are tested, it is preferable to use higher calcination temperatures (i.e., 550 °C).

It must be indicated that the catalyst heat-treated at 550 °C presented excellent stability and a reasonable robustness. This catalyst was also tested using mixtures of alkanes. Then, at a low reaction temperature (235 °C), the propane was converted into CO_2_ and was not affected by the presence of methane, probably due to the low adsorption of methane at that temperature. Unfortunately, at higher temperatures (350 °C), the presence of propane led to an important drop in the methane conversion, and this was likely due to the adsorption and reaction of propane into the cobalt oxide surface sites, thus decreasing the amount of available active sites for methane activation.

## 4. Conclusions

Cobalt oxide synthesized by a simple method involving the use of oxalic acid were highly active in the alkane total oxidation, with a performance in the order or even higher than typical catalysts based on noble metals. The calcination temperature was a determining factor in controlling the maximization of the reactivity of cobalt oxide catalysts. Low calcination temperatures are preferred over higher calcination temperatures, since it results in higher activity per both mass of catalyst and per m^2^. However, if the alkane to be eliminated presents low reactivity, as in the case of methane, intermediate calcination temperatures are optimal, otherwise the temperature to completely remove methane exceeds that of the calcination of the catalyst. The catalytic performance was related to the physicochemical characteristics of the cobalt oxides. Thus, the concentration of S1 low-coordinated oxygen species at the near surface of the Co_3_O_4_-based catalysts seemed to be related with the higher reducibility of the cobalt oxides and, consequently, with the higher areal rate. The effect of the mixture of different alkanes and the presence of steam were also studied.

## Figures and Tables

**Figure 1 materials-14-07120-f001:**
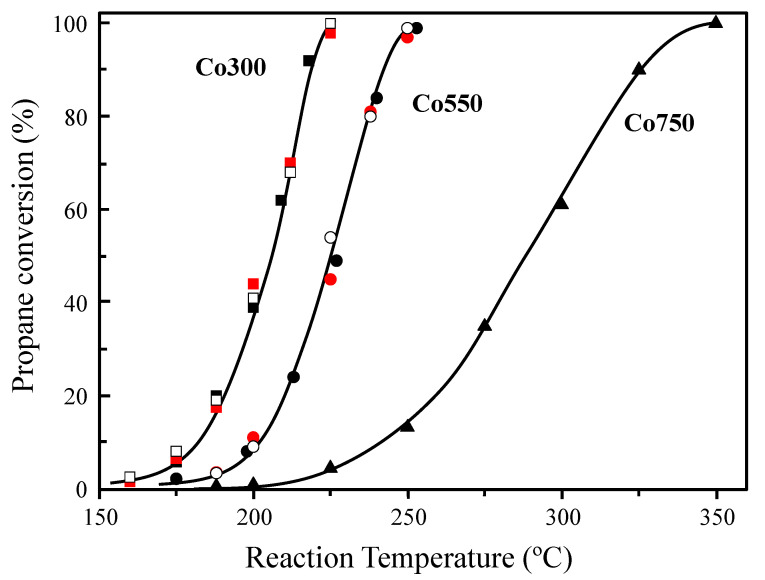
Variations in propane conversion with the reaction temperature in total oxidation of propane for consecutive catalytic cycles over Co_3_O_4_-based catalysts activated at different calcination temperatures: 300 °C (squares), 550 °C (circles), and 750 °C (triangles). First cycle (full black), second cycle (full red), and third cycle (white); reaction conditions detailed in text.

**Figure 2 materials-14-07120-f002:**
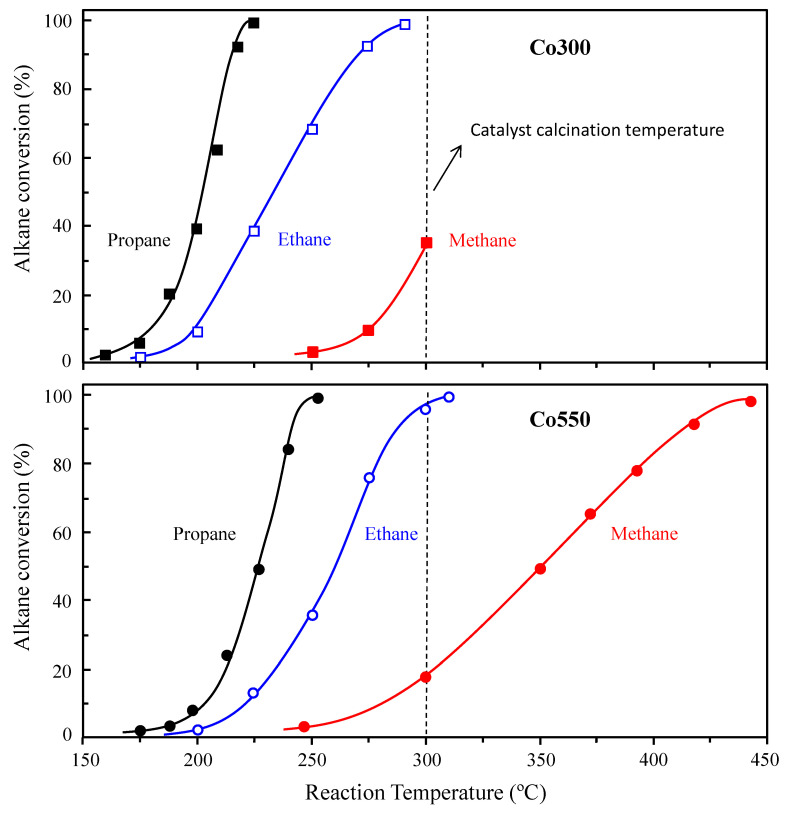
Variations in the alkane (i.e., methane, ethane, or propane) conversion with the reaction temperature in total oxidation catalytic reactions using Co_3_O_4_-based catalysts activated at different calcination temperatures: 300 °C (Co300 catalyst, up) or 550 °C (Co550 catalyst, down). Reaction conditions detailed in the text. Propane (full black), ethane (empty), and methane (full red).

**Figure 3 materials-14-07120-f003:**
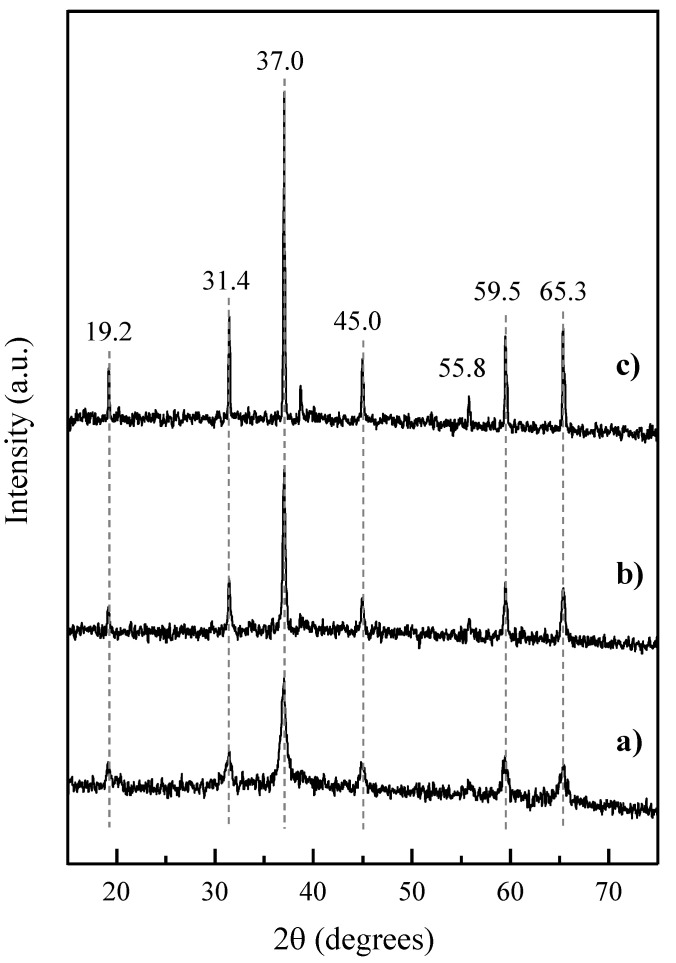
Powder X-ray diffraction (XRD) patterns of Co_3_O_4_-based catalysts activated at different calcination temperature: 300 (**a**), 550 (**b**), and 750 °C (**c**).

**Figure 4 materials-14-07120-f004:**
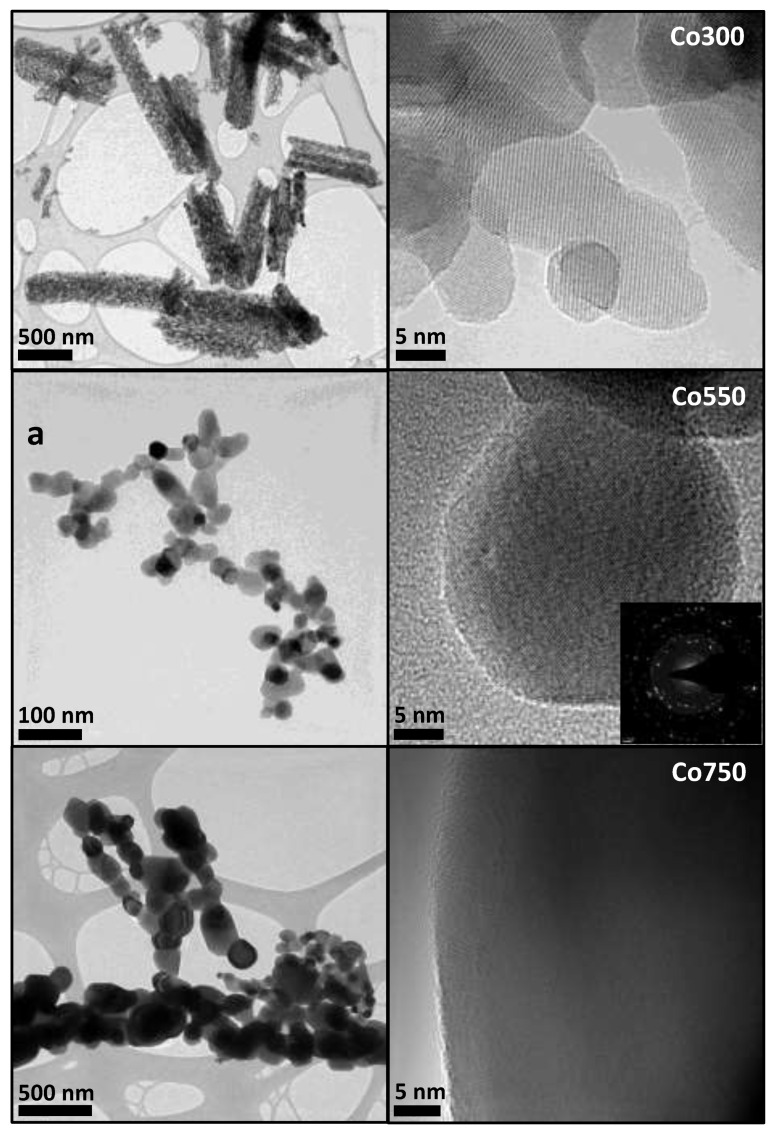
High-resolution transmission electron micrographs (HRTEMs) at lower (**left**) and higher (**right**) magnification of the Co_3_O_4_ catalysts calcined at different temperatures: 300 °C (Co300), 550 °C (Co550), and 750 °C (Co750).

**Figure 5 materials-14-07120-f005:**
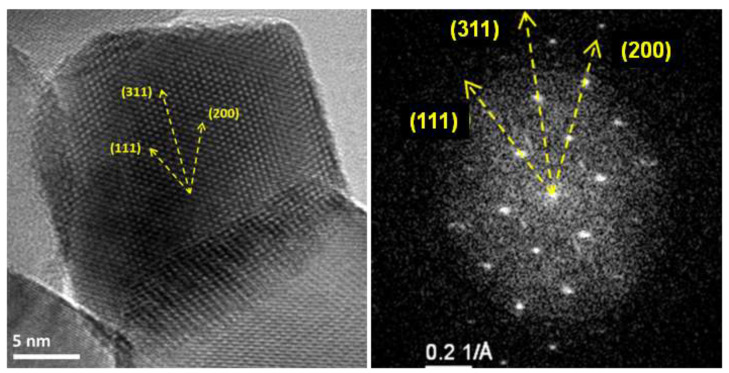
HRTEM image of the Co550 catalyst with a central Co_3_O_4_ single nanoparticle (**left**) and its corresponding fast Fourier transform (**right**) with both displaying the different crystal directions observed.

**Figure 6 materials-14-07120-f006:**
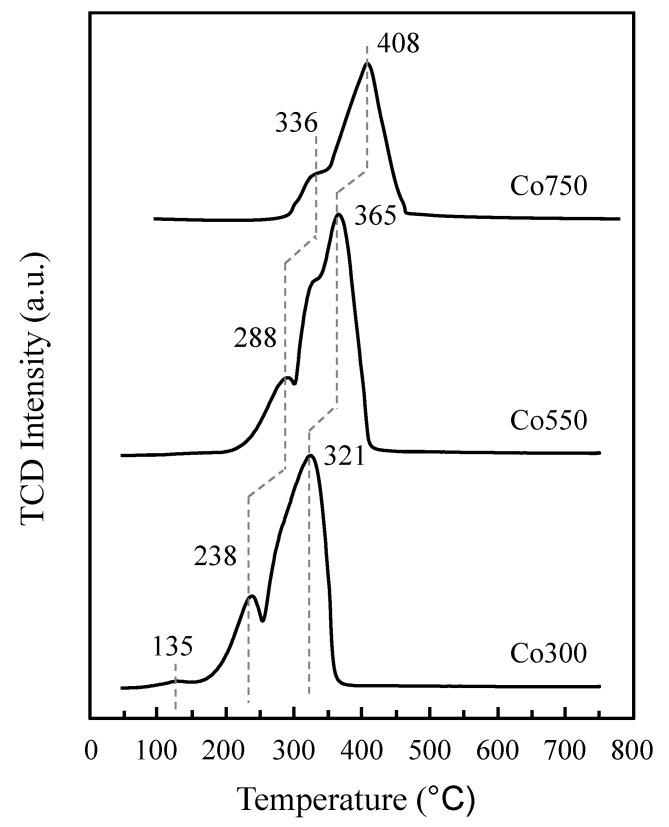
Profiles of temperature-programmed reduction with H_2_ (TPR-H_2_) collected on Co300, Co550, and Co750 catalysts. Conditions: 50 mL min^−1^ 10% H_2_/Ar flow.

**Figure 7 materials-14-07120-f007:**
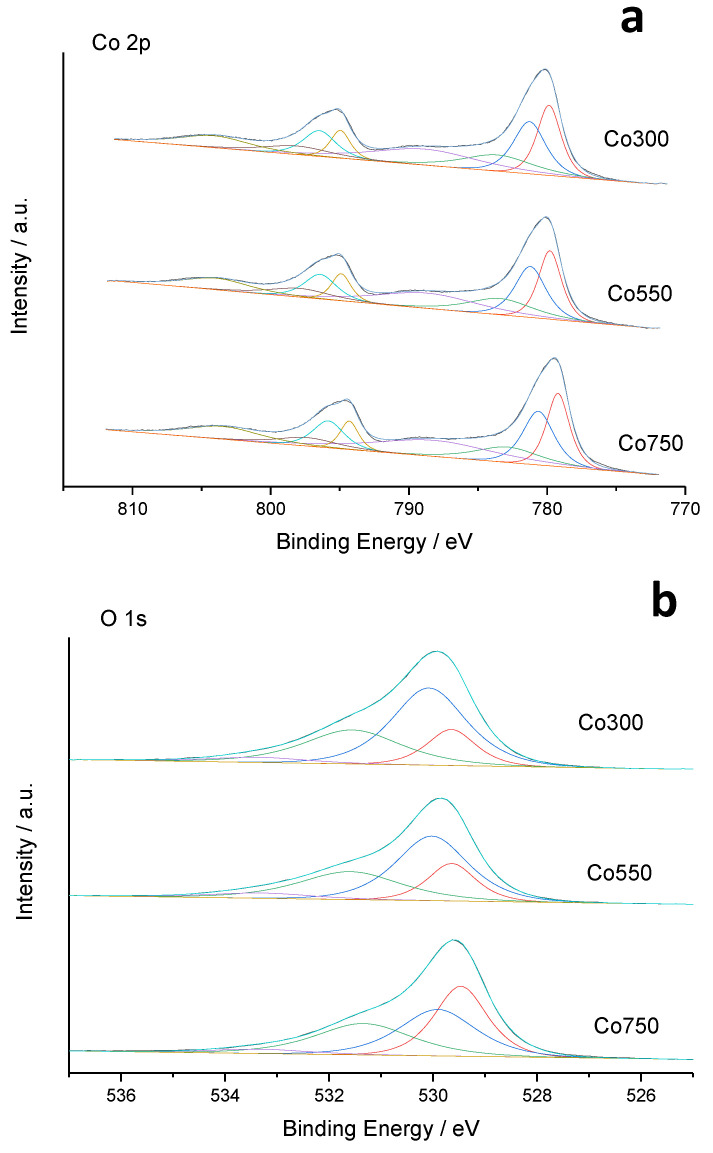
Curve fitting analysis of XPS Co(2p) (**a**) and O 1 s (**b**) core levels for cobalt oxide catalysts calcined at different temperatures.

**Figure 8 materials-14-07120-f008:**
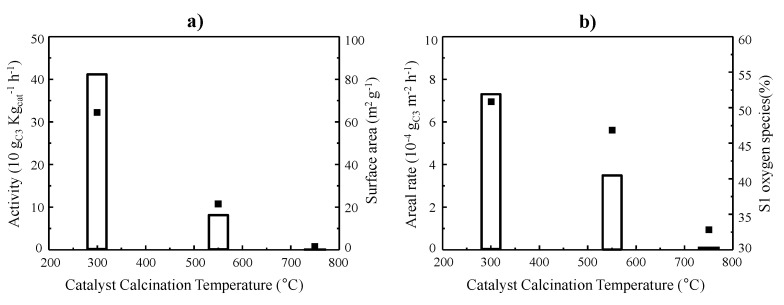
Variations in the catalytic activity per mass unit (white bars) and the BET surface area (black squares) (**a**) and variations in the areal rate (white bars) and the amount of S1 oxygen species at the near surface (black squares) (**b**) with the calcination catalysts for Co300, Co550, and Co750 catalysts. Reaction temperature = 190 °C; remaining conditions provided in the text.

**Figure 9 materials-14-07120-f009:**
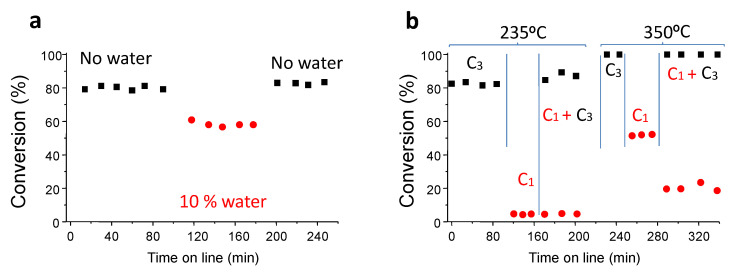
Variations in the propane conversion with the time on line in the absence (▪) or in the presence of water (●) using Co550 catalyst (**a**). Variations in the propane (▪) or methane (●) conversion with the time on line using Co550 catalyst (**b**). Reaction conditions: (**a**) reaction temperature = 235 °C, C_3_/O_2_/He = 0.8/19.8/79.4 without water or C_3_/O_2_/He/H_2_O = 0.8/17.8/71.4/10 with water; (**b**) first and forth stretch: C_3_/O_2_/He = 0.8/19.8/79.4; second and fifth stretch C_1_/O_2_/He = 0.8/19.8/79.4; third and sixth stretch C_1_/C_3_/O_2_/He = 0.8/0.8/19.7/78.7.

**Table 1 materials-14-07120-t001:** Main characteristics of the catalysts based on cobalt mixed oxide.

Catalyst	Calcination	Surface Area,	Mean Crystallite	Temperature (°C) for 20% Conversion ^b^		
	Temperature, °C	m^2^ g^−1^	Size ^a^, nm	Methane	Ethane	Propane
Co300	300	65.0	19.9 ± 0.8	285	210	190
Co550	550	22.1	32.4 ± 1.1	305	235	210
Co750	750	2.1	56.5 ± 3.2	n.d.	n.d.	265

^a^ Mean Co_3_O_4_ crystallite diameter (with standard deviation) determined from the XRD data using the Scherrer equation; ^b^ reaction conditions in the experimental section (n.d.: not determined).

**Table 2 materials-14-07120-t002:** XPS results of Co_3_O_4_-based catalysts from the curve fitting analysis of Co2p and O1s core levels.

Sample	Co^2+^/Co^3+^	Oα (%)		Oβ (%)		
		S0	S1	S2	S3	Overall Oβ
Co300	0.98	16	51	29	4	84
Co550	0.97	19	47	29	5	81
Co750	0.99	34	33	29	4	66

## Data Availability

Not applicable.

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
