# Peer review of "Highly Active Co3O4-Based Catalysts for Total Oxidation of Light C1–C3 Alkanes Prepared by a Simple Soft Chemistry Method: Effect of the Heat-Treatment Temperature and Mixture of Alkanes"

_materials, 2021, doi:10.3390/ma14237120_

Round 1
Reviewer 1 Report
This work investigated the effect of calcination temperature on Co3O4 catalysts for total oxidation of propane, ethane, and methane. Lower calcination temperature resulted in smaller particle size, larger surface area, better reducibility and more surface oxygen, and thus higher activity. The shorter the carbon chain, the harder it is to be oxidized. Methane oxidation can be hindered by the presence of propane over Co-550 catalyst. The work is not so novelty but quite carefully written. It can be published after addressing some questions.
- In the Introduction part, some recent publications on total oxidation of propane over Co3O4 catalysts were overlooked, e.g., 10.1016/j.apcatb.2020.119657, 10.1021/acscatal.1c01666, 10.1016/j.apcatb.2020.119566
- In the preparation part line 117, the authors use an oxalic acid/Co molar ratio of 3, why? As we know, an oxalic acid/Co molar ratio slightly greater than 1 is enough to precipitate Co2+, has the authors investigated the effect of oxalic acid amount? If not, I do not understand this choice that waste chemicals.
- Line 243, “JCPDS: 42-167” can be a mistake because it corresponds to yttrium silicate (Y2Si2O7), I guess it should be “JCPDS: 42-1467”.
Line 278, “JCPDS card 015-0870” corresponds to iridium oxide (IrO2), not Co3O4.
Line 278, “figure not shown here”, If possible, I suggest putting it in the supplementary materials.
- Line 329, I do not think the peak at 138 ºC can be caused by cobalt nitrate impurities since the catalyst was calcined at 300 ºC, in my opinion, this peak may result from surface adsorbed oxygen species.
- The conclusion part is too long. It seems more like a discussion than a conclusion. It should be moved to discussion part. The authors should draw a new and true conclusion.
- Some small points,
line 61 and 442, “Volatile Organic Compound”, abbreviated VOC is enough.
Line 149, “-50 and 800 °C” should be “50 and 800 °C”
Table 2, “Oα (eV)” and “Oβ (eV)” should be “Oα (%)” and “Oβ (%)”
Author Response
This work investigated the effect of calcination temperature on Co3O4 catalysts for total oxidation of propane, ethane, and methane. Lower calcination temperature resulted in smaller particle size, larger surface area, better reducibility and more surface oxygen, and thus higher activity. The shorter the carbon chain, the harder it is to be oxidized. Methane oxidation can be hindered by the presence of propane over Co-550 catalyst. The work is not so novelty but quite carefully written. It can be published after addressing some questions.
- In the Introduction part, some recent publications on total oxidation of propane over Co3O4 catalysts were overlooked, e.g., 10.1016/j.apcatb.2020.119657, 10.1021/acscatal.1c01666, 10.1016/j.apcatb.2020.119566
Response: Thanks to the reviewer for the updating of the literature. Accordingly, we have added 13 new references among which these three articles (10.1016/j.apcatb.2020.119657, 10.1021/acscatal.1c01666, 10.1016/j.apcatb.2020.119566) have been included.
- In the preparation part line 117, the authors use an oxalic acid/Co molar ratio of 3, why? As we know, an oxalic acid/Co molar ratio slightly greater than 1 is enough to precipitate Co2+, has the authors investigated the effect of oxalic acid amount? If not, I do not understand this choice that waste chemicals.
Response: We employed a large excess of oxalic acid (3 to 1 ratio) because we wanted to be sure that all the Co2+ was precipitated. Apart from that, oxalic acid exerts as a swelling agent in a way that the surface area increases. As the referee suggests, we could have used a smaller excess. Unfortunately, we have not investigated the effect of the amount of oxalic acid.
Line 243, “JCPDS: 42-167” can be a mistake because it corresponds to yttrium silicate (Y2Si2O7), I guess it should be “JCPDS: 42-1467”. Line 278, “JCPDS card 015-0870” corresponds to iridium oxide (IrO2), not Co3O4.
Response: Apologies for the mistakes. Now, we have corrected both errors.
Line 278, “figure not shown here”, If possible, I suggest putting it in the supplementary materials.
Response: According to the comment of the referee that image has been included in the manuscript. We have added it as an inset of Figure 4.
Line 329, I do not think the peak at 138 ºC can be caused by cobalt nitrate impurities since the catalyst was calcined at 300 ºC, in my opinion, this peak may result from surface adsorbed oxygen species.
Response: The interpretation given by the reviewer makes sense. Both species (impurities of the salt and adsorbed oxygen species) can be reduced at low temperatures and give that low intensity signal. We have included a sentence in that way
“The presence of surface adsorbed oxygen species could be also related to this low temperature reduction peak.”
The conclusion part is too long. It seems more like a discussion than a conclusion. It should be moved to discussion part. The authors should draw a new and true conclusion.
Response: According to the comment of the reviewer we have moved all the Conclusion section to Discussion and re-written a new shorter Conclusion section.
New Conclusions:
Cobalt oxide synthesized by a simple method involving the use of oxalic acid are highly active in the alkane total oxidation, with a performance in the order or even higher than typical catalysts based on noble metals. The calcination temperature is a determining factor to be controlled to maximize the reactivity of cobalt oxide catalysts. Low calcination temperatures are preferred over higher calcination temperatures since it results in higher activity per both mass of catalyst and per m2. However, if the alkane to be eliminated presents low reactivity, as in the case of methane, intermediate calcination temperatures are the optimal, otherwise the temperature to completely remove methane exceeds that of the calcination of the catalyst. The catalytic performance has been related to the physicochemical characteristics of the cobalt oxides. Thus, the concentration of S1 low coordinated oxygen species in the near surface of the Co3O4 based catalysts seems to be related with the higher reducibility of the cobalt oxides and consequently with the higher areal rate.
Some small points,
line 61 and 442, “Volatile Organic Compound”, abbreviated VOC is enough.
Response: Changed
Line 149, “-50 and 800 °C” should be “50 and 800 °C”
Response: Apologies for the mistake. It has now been changed
Table 2, “Oα (eV)” and “Oβ (eV)” should be “Oα (%)” and “Oβ (%)”
Response: Apologies for the mistake. It has now been changed
Reviewer 2 Report
Materials
Manuscript ID: materials-1456310
Title: Highly Active Co3O4 Based Catalysts for Total Oxidation of Light C1-C3 Alkanes prepared by a simple soft chemistry method: effect of the heat treatment temperature and mixture of alkanes
In this manuscript, the authors prepared cobalt mixed oxide (Co3O4), materials which have shown a remarkably high activity in the reaction of total oxidation of VOCs. The catalytic activity of Co3O4 catalysts has been found to be as high as that of the most effective catalysts based on noble metals. The physicochemical properties of the prepared catalysts were characterized using XRD, TPR, TPD-O2, TEM and XPS. I think that is a very interesting experimental study. I recommend a publication of this work to Materials after the authors consider the following major revisions.
Comment #1
The authors need to improve their reference list as they have used only 45 references. I think that they should aim for at least 70 references in order to make their introduction and discussion section more comprehensive. In the “Correlation between catalytic performance and physicochemical features” section, the authors should correlate their catalytic activity and stability results with the already published studies of different researchers to show the importance of their research study.
Comment #2
How did the authors determine that the catalytic samples contained the desired concentration of Co on the catalyst surface?
Comment #3
I think that it would be useful for the authors to investigate the catalytic stability for more than 6 hr (I recommend 20 hr). Then, the catalytic samples should be characterized by high resolution transmission electron microscopy (HRTEM) to check if Co3O4 remain stable in the structure. Finally, possible agglomeration should be investigated.

Author Response
Referee 2
In this manuscript, the authors prepared cobalt mixed oxide (Co3O4), materials which have shown a remarkably high activity in the reaction of total oxidation of VOCs. The catalytic activity of Co3O4 catalysts has been found to be as high as that of the most effective catalysts based on noble metals. The physicochemical properties of the prepared catalysts were characterized using XRD, TPR, TPD-O2, TEM and XPS. I think that is a very interesting experimental study. I recommend a publication of this work to Materials after the authors consider the following major revisions.
Response: Thanks for the positive comments
Comment #1
The authors need to improve their reference list as they have used only 45 references. I think that they should aim for at least 70 references in order to make their introduction and discussion section more comprehensive. In the “Correlation between catalytic performance and physicochemical features” section, the authors should correlate their catalytic activity and stability results with the already published studies of different researchers to show the importance of their research study.
Response: According to the reviewer comments we have included 13 new references. Moreover, we have discussed the results of previously published articles and compared with our work. Our results are in the order of the best results for the catalytic transformation of alkanes. We have added the following text:
The catalytic results obtained in the present study for the total oxidation of alkanes are in the same order of those obtained by the best cobalt oxide catalysts reported in the literature. Then, in propane total oxidation, to reach conversion of 90 %, a reaction tem-perature of ca. 215 oC was required with our optimal catalyst. This result is similar to that achieved by an ordered high surface area Co3O4 synthesized by a nanocasting route [20]. In other recent studies, the influence of the morphology (and its relationship with other physicochemical properties) over the catalytic activity was studied. Using rods of Co3O4, a propane conversion of 90% was obtained at 195 ºC [40]. In a similar study, Co3O4 was prepared through a solid state method to reach a propane conversion of 50 %, at reaction temperatures higher than 210 ºC [39]. In another work, it was demonstrated that the cata-lytic activity of Co3O4 catalysts was facet‐dependent, reaching 90 % conversion at 239oC in the best cases [25]. The effect of the pH in the synthesis of cobalt oxide has been also stud-ied on the catalytic performance in propane total oxidation. Then, it has been reported a conversion of ca. 90 % at 220oC, with a Co3O4 catalyst synthesized using high pH [31].
On the other hand, in the total oxidation of methane to obtain conversions of 10 or 90 % with Co550 in this study, it was required temperatures of ca. 280 or 410 ºC, respectively. Similarly, using an optimized N-doped Co3O4 catalyst 10 % methane conversion was achieved at 300 ºC and to obtain a conversion of 90 %, 400 ºC were required [59].
Comment #2
How did the authors determine that the catalytic samples contained the desired concentration of Co on the catalyst surface?
Response: Thanks for the comment. We do not know if the amount of cobalt in the surface is the optimal one although we know that Co3O4 phase, which is present in all our catalysts, is the most efficient CoOx for this reaction, as it has been previously proposed in the literature. In this case as we only have cobalt oxide (determined by HR-TEM an by XRD) with a Co3O4 formula, the theoretical Co2+/Co3+ ratio of bulk should be ca. 0.5. This data is corroborated by the TPR experiments in which a hydrogen consumption very close to the theoretical is observed in all cases. For the amount of each cobalt species on the surface we have deconvoluted the Co-signal in the XPS spectra and observed that the amount of Co2+ (ca. 1) is higher than that of the bulk Co3O4 (ca. 0.5), this suggesting the existence of oxygen vacancies.
Comment #3
I think that it would be useful for the authors to investigate the catalytic stability for more than 6 hr (I recommend 20 hr). Then, the catalytic samples should be characterized by high resolution transmission electron microscopy (HRTEM) to check if Co3O4 remain stable in the structure. Finally, possible agglomeration should be investigated.
Response: According to the comment of the reviewer we show below the results of a stability test (74 h) undertaken with the Co550 catalyst. We have observed a highly stable performance, likely due to the high difference between the reaction temperature employed (lower than 250ºC) and the calcination temperature (550ºC). This catalyst has been also characterized by HR-TEM, and a negligible agglomeration has been observed compared to the fresh catalyst, which is in agreement with the negligible deactivation observed.
Evolution of the propane conversion with the time on line on Co550 catalyst. Reaction Temperature = 225ºC and remaining reaction conditions as usual.
TEM image of Co550 after a catalytic test at 225ºC for ca. 75 h.
We have added the following text to the manuscript
“A stability test was performed with this Co550 sample showing that at a reaction temperature of 225oC the propane conversion initially observed was maintained for more than 24 h. The catalyst after use was characterized by TEM, showing no appreciable difference in terms of crystallite size compared with the fresh sample. Next, we decided to study the robustness of the Co-550 catalyst, by undertaking experiments with steam and, also, mixing hydrocarbons (Figure 9).”

Round 2
Reviewer 2 Report
Accept in present form